# Preoperative CA125 Significantly Improves Risk Stratification in High-Grade Endometrial Cancer

**DOI:** 10.3390/cancers15092605

**Published:** 2023-05-04

**Authors:** Marike S. Lombaers, Karlijn M. C. Cornel, Nicole C. M. Visser, Johan Bulten, Heidi V. N. Küsters-Vandevelde, Frédéric Amant, Dorry Boll, Peter Bronsert, Eva Colas, Peggy M. A. J. Geomini, Antonio Gil-Moreno, Dennis van Hamont, Jutta Huvila, Camilla Krakstad, Arjan A. Kraayenbrink, Martin Koskas, Gemma Mancebo, Xavier Matías-Guiu, Huy Ngo, Brenda M. Pijlman, Maria Caroline Vos, Vit Weinberger, Marc P. L. M. Snijders, Sebastiaan W. van Koeverden, Ingfrid S. Haldorsen, Casper Reijnen, Johanna M. A. Pijnenborg

**Affiliations:** 1Department of Obstetrics and Gynaecology, Radboud University Medical Center, 6525 GA Nijmegen, The Netherlands; 2Radboud Institute of Health Sciences, 6525 GA Nijmegen, The Netherlands; 3Department of Obstetrics and Gynecology, Division Gynecologic Oncology, University of Toronto, Toronto, ON M5G 1E2, Canada; 4Department of Pathology, Eurofins PAMM, 5623 EJ Eindhoven, The Netherlands; 5Department of Pathology, Radboud University Medical Center, 6525 GA Nijmegen, The Netherlands; 6Department of Pathology, Canisius-Wilhelmina Hospital, 6532 SZ Nijmegen, The Netherlands; 7Department of Oncology, KU Leuven, 3000 Leuven, Belgium; 8Center for Gynecologic Oncology Amsterdam, Netherlands Cancer Institute and Amsterdam University Medical Center, 1066 CX Amsterdam, The Netherlands; 9Department of Gynecology, Catharina Hospital, 5623 EJ Eindhoven, The Netherlands; 10Institute of Pathology, University Medical Center, 79104 Freiburg, Germany; 11Biomedical Research Group in Gynecology, Vall Hebron Institute of Research, Universitat Autònoma de Barcelona, Centro de Investigación Biomédica en Red Cáncer, 08193 Barcelona, Spain; 12Department of Obstetrics and Gynecology, Maxima Medical Centre, 5631 BM Veldhoven, The Netherlands; 13Department of Gynecology, Vall Hebron University Hospital, Centro de Investigación Biomédica en Red Cáncer, 08035 Barcelona, Spain; 14Department of Obstetrics and Gynecology, Amphia Hospital, Breda, 4818 CK Breda, The Netherlands; 15Department of Pathology, University of Turku, 20500 Turku, Finland; 16Department of Obstetrics and Gynecology, Haukeland University Hospital, 5021 Bergen, Norway; 17Centre for Cancer Biomarkers, Department of Clinical Science, University of Bergen, 5020 Bergen, Norway; 18Department of Obstetrics and Gynaecology, Rijnstate Hospital, 6815 AD Arnhem, The Netherlands; 19Department of Obstetrics and Gynecology, Bichat-Claude Bernard Hospital, 75018 Paris, France; 20Department of Obstetrics and Gynecology, Hospital del Mar, Parc de Salut Mar, 08003 Barcelona, Spain; 21Department of Pathology and Molecular Genetics and Research Laboratory, Hospital Universitari Arnau de Vilanova, University of Lleida, IRBLleida, Centro de Investigación Biomédica en Red Cáncer, 25003 Lleida, Spain; 22Department of Obstetrics and Gynecology, Elkerliek Hospital, 5751 CB Helmond, The Netherlands; 23Department of Obstetrics and Gynecology, Jeroen Bosch Hospital, 5223 GZ ‘s-Hertogenbosch, The Netherlands; 24Department of Obstetrics and Gynecology, Elisabeth-TweeSteden Hospital, 5000 LC Tilburg, The Netherlands; 25Department of Gynecology and Obstetrics, University Hospital Brno, Faculty of Medicine, Masaryk University, 601 77 Brno, Czech Republic; 26Department of Obstetrics and Gynecology, Canisius-Wilhelmina Hospital, 6532 SZ Nijmegen, The Netherlands; 27Department of Radiology and Nuclear Medicine, Radboud University Medical Center, 6525 GA Nijmegen, The Netherlands; 28Mohn Medical Imaging and Visualization Centre, Department of Radiology, Haukeland University Hospital, 5021 Bergen, Norway; 29Department of Radiation Oncology, Radboud University Medical Center, 6525 GA Nijmegen, The Netherlands

**Keywords:** endometrial cancer, advanced stage, outcome, high-grade, CA125

## Abstract

**Simple Summary:**

Patients with high-grade uterine cancer (UC) have a risk of around 20% of the cancer spreading to the lymph nodes, while this is only around 10% in patients with low-grade uterine cancer. CA125 is a marker that can be detected in blood and is associated with increased tumor spread. Studies on CA125 and its association with tumor spread within low-grade UC exist but are limited for high-grade UC. The primary aim of this retrospective study was to assess whether elevated CA125 is predictive for UC spread and survival. Secondarily, we studied the additional value of preoperative imaging by CT scan in relation to CA125 specifically in high-grade UC. We observed that elevated CA125 was related to advanced stage and LNM in high-grade UC and a worse prognosis. If CA125 was normal, the additional value of CT to predict lymph node spread was limited.

**Abstract:**

Patients with high-grade endometrial carcinoma (EC) have an increased risk of tumor spread and lymph node metastasis (LNM). Preoperative imaging and CA125 can be used in work-up. As data on cancer antigen 125 (CA125) in high-grade EC are limited, we aimed to study primarily the predictive value of CA125, and secondarily the contributive value of computed tomography (CT) for advanced stage and LNM. Patients with high-grade EC (n = 333) and available preoperative CA125 were included retrospectively. The association of CA125 and CT findings with LNM was analyzed by logistic regression. Elevated CA125 ((>35 U/mL), (35.2% (68/193)) was significantly associated with stage III-IV disease (60.3% (41/68)) compared with normal CA125 (20.8% (26/125), [*p* < 0.001]), and with reduced disease-specific—(DSS) (*p* < 0.001) and overall survival (OS) (*p* < 0.001). The overall accuracy of predicting LNM by CT resulted in an area under the curve (AUC) of 0.623 (*p* < 0.001) independent of CA125. Stratification by CA125 resulted in an AUC of 0.484 (normal), and 0.660 (elevated). In multivariate analysis elevated CA125, non-endometrioid histology, pathological deep myometrial invasion ≥50%, and cervical involvement were significant predictors of LNM, whereas suspected LNM on CT was not. This shows that elevated CA125 is a relevant independent predictor of advanced stage and outcome specifically in high-grade EC.

## 1. Introduction

Endometrial carcinoma (EC) is the most common gynecological malignancy in industrialized countries. Primarily, distinction on outcome is based on tumor grade, with favorable outcomes in low-grade tumors and poor outcomes in high-grade tumors. In high-grade EC, i.e., grade 3 endometrioid and non-endometrioid histology, there is an increased risk of advanced stage and lymph node metastasis (LNM) [1]. As the risk of LNM in grade 3 EC varies between 15–44% depending on the histological subtype and myometrial invasion (MI), determination of lymph node status by lymphadenectomy or sentinel node (SN) biopsy is recommended in patients without clinical suspicion of advanced stage EC [2,3,4,5]. In the preoperative work-up abdominal computed tomography (CT), pelvic magnetic resonance imaging (MRI), and 18FDG positron emission tomography (PET)-CT can be considered to detect extra-uterine tumor spread or distant metastases, as this may impact the surgical approach [2,3,6,7]. After primary surgico-pathological staging, information about tumor stage, histopathological subtype, tumor grade, presence of deep myometrial invasion ≥50% (DMI), cervical stromal invasion (CI), lymphovascular space invasion (LVSI), or LNM guides adjuvant radiotherapy and/or chemotherapy [2,3]. The recently introduced molecular classification may increasingly guide adjuvant therapy, yet for high-grade EC surgical staging is still recommended according to the ESGO-ESTRO-ESP guideline [2]. In a systematic review and meta-analysis, it was shown that elevated cancer antigen 125 (CA125) serum level is associated with increased risk of LNM both in low-grade and high-grade ECs [8]. In addition, CA125 has been incorporated in several predictive models that have shown an improved risk classification for advanced stage and LNM in EC and remains an independent prediction compared to molecular classification [9,10,11,12,13,14]. Furthermore, imaging findings indicating extra-uterine spread and/or distant spread predict advanced stage and poor prognosis [6,8]. However, preoperative CT and MRI have limited sensitivity for the identification of LNM [6,8,13,15,16]. So far, preoperative CA125 has not been selectively studied in patients with high-grade EC in relation to imaging findings. Therefore, our primary aim is to determine the predictive value of CA125 in relation to LNM and advanced stage in high-grade EC. Our secondary aim is to determine the predictive value of CA125 combined with preoperative imaging in relation to LNM and advanced stage in high-grade EC.

## 2. Materials and Methods

### 2.1. Patient Cohort

A retrospective multicenter study was performed including patients with preoperative high-grade EC with all histological subtypes, diagnosed by endometrial sampling by pipelle, dilatation and curettage, or hysteroscopic biopsy. Patients were retrieved from three well-documented study cohorts [13,17,18,19]. Patients who underwent surgical staging but with either preoperative or postoperative grade 3 tumors, remained included for analysis. Patients without available preoperative CA125 were excluded. In all study cohorts, histopathological analysis was performed by gynecological pathologists. Approval for the original studies was obtained by the Review Board Radboud University Medical Center Nijmegen, the Netherlands (institutional study protocol 2015-2101) and by the medical ethical committee of the Elisabeth-Tweesteden Hospital Tilburg, the Netherlands (protocol 1129).

### 2.2. Data Collection

Collected data consisted of patient characteristics (age, body mass index [BMI], comorbidity), preoperative CA125, modality and results of preoperative imaging as reported in the routine radiology report, type of surgical procedure, pathological lymph node status, histology type, and stage of disease. Lymphadenopathy on imaging was defined as lymph nodes with short axis diameter ≥10 mm and with or without suspected distant spread on imaging [20]. Scans were read and reported by radiologists at the hospital where the patient was treated. When imaging findings reported in the radiology report were ‘inconclusive’ for lymph node status, they were excluded from analysis. Surgical lymph node staging was defined as pelvic and/or para-aortic lymph node sampling, with or without omental sampling, peritoneal biopsies, or both. Both preoperative and postoperative histopathological data on tumor histology and grade were documented. Patients who underwent staging with distant metastases but without LNM were excluded. Due to a limited number of performed MRIs (n = 14), myometrial invasion (MI) and cervical stromal invasion (CI) were documented from the postoperative pathology report only. When CI findings did not specify whether there was stromal or only endocervical invasion, they were excluded from analysis. The presence of LNM, LVSI and the final FIGO stage were documented from the final postoperative pathology report after surgical staging. Adjuvant treatment was classified into: none, radiotherapy including external beam radiation (EBRT) with or without vaginal brachytherapy (VBT), VBT only, chemotherapy, and combined chemoradiotherapy. Follow-up data including disease-specific- and overall survival were collected for an average of 32 months after diagnosis.

### 2.3. Statistical Analysis

Clinicopathological differences between the three cohorts and subgroups were compared using Pearson’s chi-square test or Fisher’s exact test for categorical data. The Mann–Whitney U test was used for continuous data. It has been shown that different cut-off values for CA125 serum levels (25 U/mL and 35 U/mL) had comparable diagnostic accuracy for the prediction of LNM [8]. We used the cut-off value of 35 U/mL as dichotomous variable for ‘normal’ (≤35 U/mL) and ‘elevated’ (>35 U/mL) CA125, according the widely used cut-off value in hospital laboratories [11,21,22]. Kaplan–Meier curves were created for 5-year disease-specific survival (DSS) and overall survival (OS) for all patients within the cohort with available data on recurrence and death. Patients with progression of disease were excluded from DSS analysis. A log rank (Mantel–Cox) test was run to compare DSS and OS in groups with and without elevated CA125. For the analysis on LNM and advanced International Federation of Gynecology and Obstetrics (FIGO) stage, defined as FIGO stage III and IV, all patients who underwent surgical staging were included. A *p* value less than 0.05 was considered significant. Sensitivity, specificity, positive predictive value (PPV), negative predictive value (NPV), and Clopper–Pearson exact confidence intervals for CA125 and CT imaging versus LNM were calculated. Receiver operating characteristics (ROC) were made to calculate the area under the curve (AUC). Age, BMI, non-EC histology, MI, and CI were identified from earlier studies as additional predictive variables for LNM and included in univariable analysis for LNM [1,8,13,23]. Significant predictive variables from univariate logistic regression analysis (*p* < 0.02) were included in a multivariate logistical regression analysis. Statistical Package for the Social Sciences (SPSS) version 25 (SPSS IBM, New York, NY, USA) software was used for data management and to perform the statistical analyses.

In accordance with the journal’s guidelines, we will provide our data for independent analysis by a selected team by the Editorial Team for the purposes of additional data analysis or for the reproducibility of this study in other centers if such is requested.

## 3. Results

### 3.1. Study Cohort

A total of 333 patients with preoperative CA125 serum level and high-grade EC were included. An overview of the inclusion of patients is shown in Appendix A. The baseline characteristics in relation to CA125 serum level are shown in Table 1, specified for patients with (n = 193) and without (n = 140) surgical lymph node (LN) staging. The median age was 66 and 72 years, respectively, and median BMI 26.8 kg/m^2^ versus 28.2 kg/m^2^, respectively. The differences in age and BMI were not statistically significant. Of all patients, 2.4% (8/333) did not undergo surgery. Overall, 44.1% (147/334) of the patients had an elevated CA125 serum level. Patients without surgical LN staging had elevated CA125 serum levels in 56.4% (79/140) of patients and presented with FIGO IV in 30.0% (42/140), whereas patients with surgical staging presented with elevated CA125 in 35.2% (68/193) and FIGO IV in 9.8% (19/193). Preoperative imaging was performed in 68.2% (n = 227) of the patients; abdominal/chest CT in 64.3% (214/333) and pelvic MRI in 4.2% (14/333). In one patient, both CT and MRI were performed. According to the radiology report, LNM were suspected in 16.3% (37/227) of patients and extra-uterine or distant spread in 8.8% (20/228). Both LNM and extra-uterine or distant spread were reported in 1.8% (4/227) of the patients. Patients with preoperative imaging underwent surgical LN staging in 67.8% (154/227) compared to 34.0% (14/40) in patients without preoperative imaging (*p* < 0.001). Excluded patients (n = 185) due to lack of CA125 did not differ with respect to baseline characteristics from the included patients.

### 3.2. CA125 in Relation to Extra-Uterine Disease and LNM

Surgically staged patients with elevated CA125 were significantly more often diagnosed with FIGO stage III-IV (60.3% [41/68]) compared to patients with normal CA125 (20.8% [26/125], *p* < 0.001). In addition, in patients with elevated CA125 the prevalence was significantly higher for DMI, CI and LVSI (Table 1). In surgically staged patients (n = 194), LNM were present in 29.4% (57/194), respectively in 56.5% (39/69) of the patients with elevated CA125 and in 14.4% (18/125) of patients with normal CA125 (*p* < 0.001). The specificity, sensitivity, NPV, PPV, and AUC of CA125 > 35 U/mL for predicting LNM are summarized in Table 2. The AUC for CA125 in relation to pathological confirmed LNM was 0.721, and 0.675 for FIGO III/IV.

### 3.3. Imaging in Relation to Extra-Uterine Disease and LNM

For the imaging analysis, only patients who underwent preoperative imaging by CT with conclusive results (n = 134) were included, as the number of patients who underwent MRI was limited (n = 10). Patients with suspected extra-uterine disease on CT (33/134), either LNM (n = 27), distant spread (n = 5), or both LNM and distant spread (n = 1) had significantly more often FIGO stage III-IV (54.5% [18/33] vs. 23.8% [24/101], *p* < 0.002). In patients with suspected extra-uterine disease on CT, the prevalence of DMI, CI, and LVSI was comparable. In patients with preoperative CT with suspected LNM (n = 28), 53.6% (15/28) had histologically confirmed LNM compared to 21.8% (22/101) in patients with no signs of LNM on CT (*p* < 0.001). The AUC for suspected LNM on CT in relation to confirmed LNM was 0.623 (Table 2) and 0.633 for FIGO III/IV.

### 3.4. CA125 and CT Results in Relation to LNM

Combined preoperative CA125 and CT results in relation to LNM are shown in Figure 1 and illustrate the relevance of CA125 in accordance with the summarized data in Table 2. Within patients with elevated CA125, CT scan with suspicion of LNM resulted in an AUC of 0.660, while with normal CA125, the AUC was 0.484. For advanced FIGO stage, similar results were observed, as the AUC for CT within patients with elevated CA125 was 0.633, while it was 0.545 in patients with normal CA125. In univariate logistic regression analysis, non-endometrioid (NEEC) histology, suspected LNM on preoperative imaging, elevated CA125, DMI and CI (from hysterectomy specimen) were significant predictive variables for LNM (Table 3). In multivariate logistic regression analysis, NEEC histology, elevated CA125 serum level, DMI, and CI remained significantly associated with LNM, whereas suspected LNM on preoperative CT was not significant.

### 3.5. CA125 in Relation to Outcome

Of all 333 patients, 331 patients had available recurrence data with a mean of 19 months until recurrence. The disease-specific survival (DSS) for patients with normal and elevated CA125 is shown in Figure 2a and was significantly different (*p* < 0.001). Endometrial cancer-related mortality data were available for 320 patients. The overall survival (OS) for patients with normal and elevated CA125 is shown in Figure 2b and was significantly different (*p* < 0.001). Both the DSS and OS remained significantly different for normal and elevated CA125 within the surgically staged patients (n = 193).

## 4. Discussion

### 4.1. Summary of Main Results

In the current study we have demonstrated that preoperative CA125 is a statistically significant predictor for advanced stage and LNM in high-grade EC. In addition, it was shown that suspected LNM on preoperative CT was a significant predictor for histologically confirmed LNM in high-grade EC but had no additional predictive value when preoperative CA125 was normal. We also demonstrated that DSS and OS are significantly worse for patients with elevated CA125. To our knowledge this is so far the largest study evaluating CA125 as prognostic biomarker specifically in patients with high-grade EC. The overall prevalence of 29.5% LNM is in line with previous studies [1,24,25]. With normal CA125, the negative predictive value for LNM was 85.6%.

Within patients with normal serum CA125 levels, preoperative CT with suspected LNM was not significant for predicting risk of LNM, and results were similar for CT and predicting advanced FIGO stage, supporting the limited additional benefit of preoperative CT imaging for staging in these patients. In the current study cohort, within patients with normal CA125, distant metastases including pulmonary metastases were present in 1.5% (2/135).

### 4.2. Results in the Context of Published Literature

The prognostic value of CA125 for advanced FIGO stage has been repeatedly shown in EC [22,26]. In a meta-analysis by Reijnen et al., both CA125 and imaging results were shown to predict the risk of LNM [8]. Within subgroup analysis of high-grade patients, preoperative CA125 resulted in an AUC of 0.745 compared to 0.638 for preoperative CT scan, both in line with our findings. Two other studies reported on the use of CA125 as predictor for LNM in high-grade EC and showed a significant association of CA125 with FIGO stage. However, only patients with either serous EC (n = 26) or clear cell EC (n = 61) histology were included without incorporation of other clinicopathological risk factors [27,28]. The low sensitivity of detection of LNM by CT is in line with previous studies and is likely attributed to the fact that the lymph nodes should be enlarged to be suspicious for metastatic on CT [20]. Although 18-FDG PET-CT scan is more sensitive than CT scan, microscopic metastasis still might be easily missed by any imaging method and require histological evaluation of the lymph nodes [29]. Multiple studies have reported on the different imaging modalities for detecting LNM in EC patients [6,20]. Both 18-FDG PET-CT and conventional CT are mostly used in clinical practice to evaluate LNM and distant metastases, with PET-CT being superior with a pooled sensitivity of 72% [29]. Our observed sensitivity of 40.5% to detect LNM on CT is in line with the aforementioned publications. The lack of association between LNM and preoperative imaging in the multivariate analysis is most likely attributed to the low sensitivity of CT. The importance of DMI and CI as independent predictors of LNM in our multivariate analysis is in line with previous studies [1,8,23]. A recent study of Fasmer et al. demonstrated how selective use of 18-FDG PET-CT scan in patients with increased risk of extended EC, based on MRI (i.e., LNM, DMI, or CI) could be incorporated. Unfortunately, CA125 was not available in this study cohort, which might be interesting to add in future refined diagnostic work-up strategies [7]. Interestingly, elevated CA125 was significantly associated with the presence of LVSI, and thus could serve as a surrogate biomarker given its association with LNM in the multivariate analysis.

While CA125 is a very sensitive marker that can be used in postmenopausal women in the diagnostic work-up for endometrial cancer, it might be less suitable in premenopausal women. This is due to the fact that elevated CA125 is seen in several benign and physiological processes such as menstrual periods, endometriosis, and pregnancy [30,31]. For patients with a wish of fertility preservation, evidence-based oncofertility counseling is an important part of the preoperative work-up [32]. So far, it is not clear whether preoperative CA125 in these young women can equally contribute to the risk stratification of endometrial cancer.

The use of MRI or expert ultrasound to determine myometrial invasion has been recommended by the recent ESGO-ESTRO-ESP guideline, but so far not yet incorporated in the SGO guideline [3,10,13,24,25,33]. Selective use of preoperative MRI, specifically in patients with high-grade and elevated CA125, could aid in improving a cost-effective surgical approach in a preoperative setting. The reported sensitivity of MRI for assessment of CI is up to 59% with a specificity up to 91% [20,34].

While this study has shown that CA125 has a good predictive value for LNM and advanced stage, its discriminatory value is not enough to forgo surgical staging based solely on CA125 with or without CT. The results do however confirm that CA125 is a valuable marker that could be used in prediction models, such as the ENDORISK model which incorporates CA125, imaging and other factors [13].

Since the introduction of molecular classification in EC, prognostication has significantly improved by allocating patients to four molecular subgroups [14,35]. These subgroups may guide tailored adjuvant therapy, but as demonstrated by Jamieson et al. could also assist in the primary risk estimation of LNM. Interestingly, it was demonstrated that in addition to the molecular subgroup, CA125 remained relevant in the multivariate analysis, which underlines the relevance of our findings even within molecular classification. In the future, other molecular and genetic markers such as non-coding RNA (ncRNA) might further stratify tumor characterization and therapeutic targeting [36,37]. Further research will be needed to assess CA125 in relation to these markers but may enable the possibility to base the need of surgical staging and/or adjuvant treatment solely on these variables.

### 4.3. Strengths and Weaknesses

Inherent to the retrospective character of our study there are limitations that need to be addressed. Overall, 41.9% (n = 140) of patients with high-grade EC were not surgically staged and thus not included in the statistical analysis for LNM, which might have introduced selection bias. Yet, as 30% of patients without surgical staging were diagnosed with FIGO IV, omitting surgical staging is inherent to standard of care. Furthermore, selection bias might have been introduced due to using preoperative tumor grade, since this represents clinical practice where patients’ treatment is based upon preoperative characteristics. However, occasionally this resulted in postoperative downgrading of EC after final pathological examination (n = 25). Based on the limited number, it is unlikely that this has significantly impacted our results. This discordance has been previously investigated in several studies who reported more extensive surgical treatment and an intermediate prognosis in patients who are ’over-graded’ compared to correctly low-graded patients [28,29,30].

### 4.4. Implications for Practice and Future Research

The clinical applicability of incorporating CA125 in the diagnostic work-up of EC has been supported by several studies [3,14,25]. The well-established strong relation between LNM and histopathological DMI and CI, supports further research of the value of preoperative pelvic MRI, especially in patients with elevated CA125 serum levels. This might further improve identification of high-risk patients and individualized surgical approach even in the era of molecular profiling.

## 5. Conclusions

Preoperative CA125 is a relevant predictive marker for advanced stage and outcome in patients with high-grade EC. While CT was a valuable predictor of LNM and advanced stage in patients with elevated CA125, it was of limited additional value in patients with normal CA125. These results support that adding CA125 can improve preoperative risk stratification specifically for patients with high-grade EC.

## Figures and Tables

**Figure 1 cancers-15-02605-f001:**
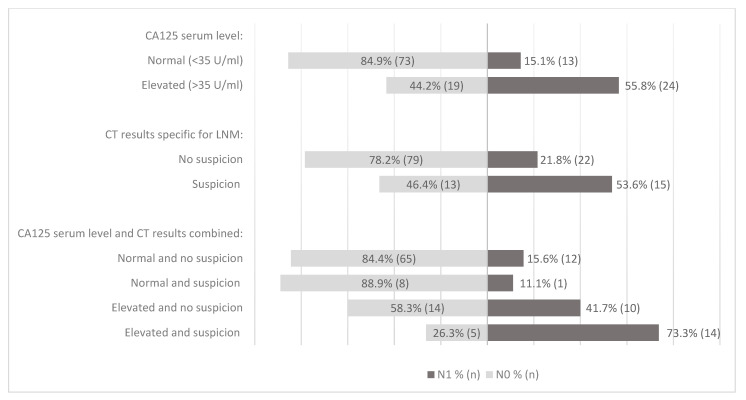
Study population proportion of LNM in relation to CA125 levels and LNM on imaging. N0: no LNM, N1: presence of LNM. CA125 = cancer antigen 125, LNM = lymph node metastases.

**Figure 2 cancers-15-02605-f002:**
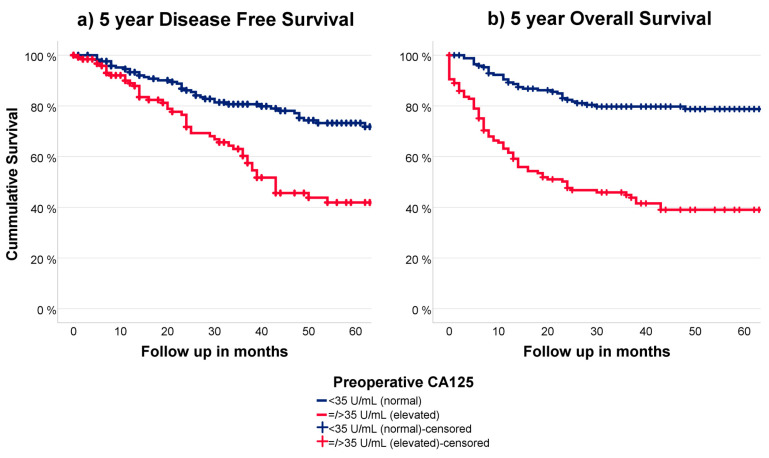
(**a**) 5 year DFS in relation to preoperative CA125 (**b**) 5 year OS in relation to preoperative CA125.

**Table 1 cancers-15-02605-t001:** Baseline characteristics of patients with preoperative high-grade EC and surgical staging versus no staging, in association with CA125 level.

			Surgical LN Staging (n = 193)				No Surgical LN Staging (n = 140)		
		Total (n = 193)	CA125 <35 U/mL (n = 125)	CA125 >35 U/mL (n = 68)	*p* Value	Total (n = 140)	CA125 <35 U/mL (n = 61)	CA125 >35 U/mL (n = 79)	*p* Value
**Age (years)**		66 (35–83)	66 (35–83)	66 (44–82)	0.477	72 (48–93)	74 (48–91)	71 (51–93)	0.234
**BMI (kg/m^2^)**		26.8 (17.6–56.0)	26.8 (17.6–56.0)	28.2 (18.0–42.6)	0.750	28.2 (16.4–49.5)	27.9 (19.5–47.8)	29.1 (16.4–49.5)	0.486
**Imaging modality**	CT	145 (86.3)	94 (84.7)	51 (89.5)	0.393	69 (69.7)	30 (55.6)	39 (86.7)	<0.001
	MRI	10 (12.0)	8 (15.4)	2 (8.0)	0.485	4 (16.0)	3 (18.8)	1 (11.1)	1
**CT results**	No extra-uterine disease	101 (69.7)	77 (81.9)	24 (47.1)	<0.001	43 (62.3)	22 (73.3)	21 (53.8)	0.052
	Suspected LNM with or without suspected distant metastasis	28 (19.3)	9 (9.6)	19 (37.3)		11 (15.9)	6 (20.0)	5 (12.8)	
	Suspected distant metastasis without suspected LNM	5 (3.4)	2 (2.1)	3 (5.9)		14 (20.3)	2 (6.7)	12 (30.8)	
	Inconclusive	11 (7.6)	6 (6.4)	5 (9.8)		1 (1.4)	0 (0.0)	1 (2.6)	
**FIGO stage ^a^**	IA	69 (35.8)	59 (47.2)	10 (14.7)	<0.001	32 (22.5)	25 (41.0)	7 (8.9)	<0.001
	IB	40 (20.7)	30 (24.0)	10 (14.7)		36 (25.7)	22 (36.1)	14 (17.7)	
	II	17 (8.8)	10 (8.0)	7 (10.3)		10 (7.1)	5 (8.2)	5 (6.3)	
	IIIA	7 (3.6)	5 (4.0)	2 (2.9)		12 (8.6)	3 (4.9)	9 (11.4)	
	IIIB	1 (0.5)	1 (0.8)	0 (0.0)		5 (3.6)	1 (1.6)	4 (5.1)	
	IIIC1-2	40 (20.7)	17 (13.6)	23 (33.8)		3 (2.1)	0 (0.0)	3 (3.8)	
	IVA	1 (0.5)	1(0.8)	0 (0.0)		7 (5.0)	1 (1.6)	6 (7.6)	
	IVB	18 (9.3)	2 (1.6)	16 (23.5)		37 (25.0)	4 (6.6)	31 (39.2)	
**Tumor grade ^a^**	1	3 (1.6)	0 (0.0)	3 (4.4)	0.020	6 (4.3)	5 (8.3)	1 (1.3)	0.040
	2	12 (6.2)	10 (8.0)	2 (2.9)		4 (2.9)	3 (5.0)	1 (1.3)	
	3	178 (92.2)	115 (92.0)	63 (92.6)		129 (92.8)	52 (86.7)	77 (97.5)	
		**Total** **(n = 193)**	**CA125** **<35 U/mL (n = 125)**	**CA125** **>35 U/mL (n = 68)**	***p* value**	**Total** **(n = 140)**	**CA125** **<35 U/mL (n = 61)**	**CA125** **>35 U/mL (n = 79)**	***p* value**
**Histology ^a^**	Endometrioid	80 (41.5)	54 (43.2)	26 (38.2)	0.480	56 (40.0)	38 (62.3)	18 (22.8)	<0.001
	Serous	89 (46.1)	53 (42.4)	36 (52.9)		74 (52.9)	19 (31.1)	55 (69.6)	
	Clear cell	9 (4.7)	7 (5.6)	2 (2.9)		1 (0.7)	1 (1.6)	0 (0.0)	
	Other	15 (7.8)	11 (8.8)	4 (5.9)		9 (6.4)	3 (4.9)	6 (7.6)	
**MI ^a^**	<50%	94 (48.7)	70 (56.0)	24 (35.3)	0.006	55 (41.7)	30 (50.0)	25 (34.7)	0.076
	=/>50%	99 (51.3)	55 (44.0)	44 (64.7)		77 (58.3)	30 (50.0)	47 (65.3)	
**CI ^a^**	No	143 (74.1)	102 (83.6)	41 (62.1)	<0.001	94 (71.8)	47 (78.3)	47 (66.2)	0.124
	Yes	45 (23.3)	20 (16.4)	25 (37.9)		37 (28.2)	13 (21.7)	24 (33.8)	
**LVSI ^a^**	No	110 (57.0)	91 (72.8)	19 (27.9)	<0.001	70 (52.6)	39 (65.0)	31 (42.5)	0.010
	Yes	83 (43.0)	34 (27.2)	49 (72.1)		63 (47.4)	21 (35.0)	42 (57.5)	
**LNM ^b^**	No (N0)	136 (70.5)	107 (85.6)	29 (42.6)	<0.001	-	-	-	-
	Yes (N1)	57 (29.5)	18 (14.4)	39 (57.4)		-	-	-	
**Adjuvant therapy**	Radiotherapy	82 (42.5)	60 (48.0)	22 (32.4)	0.006	59 (42.1)	36 (59.0)	23 (29.1)	<0.001
	Chemotherapy	60 (31.1)	28 (22.4)	32 (47.1)		32 (22.9)	4 (6.6)	28 (35.4)	
	Radio- and chemo-therapy	10 (5.2)	7 (5.6)	3 (4.4)		5 (3.6)	2 (3.3)	3 (3.8)	
	No adjuvant therapy	41 (21.2)	30 (24.0)	11 (16.2)		44 (31.4)	19 (31.1)	25 (31.6)	
**Radio-therapy**	VBT	36 (39.1)	24 (35.8)	12 (48.0)	0.287	23 (35.9)	15 (39.5)	8 (30.8)	0.476
	EBRT ± VBT	56 (60.9)	43 (64.2)	13 (52.0)		41 (64.1)	23 (60.5)	18 (69.2)	
**Follow up (months)**		39.0 (0–193)	47.5 (0–193)	33.8 (0–123)	0.003	21.0 (0–132)	36.0 (0–132)	9.0 (0–83)	<0.001
**Recurrence**	No	120 (62.5)	92 (74.2)	28 (41.2)	<0.001	64 (46.0)	41 (68.3)	23 (29.1)	<0.001
	Yes	53 (27.6)	27 (21.8)	26 (38.2)		40 (28.8)	17 (28.3)	23 (29.1)	
	Progression	19 (9.9)	5 (4.0)	14 (20.6)		35 (25.2)	2 (3.3)	33 (41.8)	
**Death**	No	125 (65.4)	94 (75.8)	31 (46.3)	<0.001	68 (49.6)	43 (70.5)	25 (32.9)	<0.001
	Yes, caused by EC	54 (28.3)	23 (18.5)	31 (46.3)		56 (40.9)	12 (19.7)	44 (57.9)	
	Yes, not caused by EC	6 (3.1)	5 (4.0)	1 (1.5)		8 (5.8)	4 (6.6)	4 (5.3)	
	Yes, unknown cause	6 (3.1)	2 (1.6)	4 (6.0)		5 (3.6)	2 (3.3)	3 (3.9)	

Values are presented as median (range) or number (%), ^a^ = based on postoperative pathology, ^b^ = based on surgical staging, LN = lymph node, CA125 = cancer antigen 125, BMI = body mass index, CT = computed tomography, MRI = magnetic resonance imaging, LNM = lymph node metastasis, FIGO = International Federation of Gynecology and Obstetrics, MI = myometrial invasion, CI= cervical stromal invasion, LVSI = lymphovascular space invasion, VBT = vaginal brachytherapy, EBRT = External Beam Radiation Therapy, EC = endometrial cancer. Missing values are not shown or used in analysis for this table.

**Table 2 cancers-15-02605-t002:** Performance of preoperative CA125 serum level and preoperative CT alone and combined versus LNM in patients who underwent surgical LN staging.

			Total (n)	Sensitivity % [95% CI] (n)	Specificity % [95% CI] (n)	PPV % [95% CI] (n)	NPV % [95% CI] (n)	ROC AUC (95% CI)	*p* Value
LNM	CA125 ^a^		193	68.4 [54.8–80.1] (39/57)	78.7 [70.8–85.2] (107/136)	57.4 [44.8–69.2] (39/68)	85.6 [78.2–91.2] (107/125)	0.721 (0.619–0.824)	<0.001
	CT ^b^		129	40.5 [24.8–7.9] (15/37)	85.9 [77.1–92.3] (79/92)	53.6 [33.9–72.5] (15/28)	78.2 [68.9–85.8] (79/101)	0.623 (0.520–0.744)	<0.001
	CA125 <35 U/mL	CT ^b^	86	7.7 [0.0–36.0] (1/13)	89.0 [79.5–95.2] (65/73)	11.1 [0.0–48.3] (1/9)	84.4 [74.4–91.7] (65/77)	0.484 (0.316–0.652)	1.000
	CA125 >35 U/mL	CT ^b^	43	58.3 [36.6–77.9] (14/24)	73.7 [48.8–90.9] (14/19)	73.7 [48.8–90.9] (14/19)	58.3 [36.6–77.9] (14/24)	0.660 (0.495–0.826)	0.036

^a^ = cut-off for positive test of 35 U/mL, ^b^ = cut-off for positive test is suspected LNM on CT for predicting LNM. CA125: cancer antigen 125, LNM: lymph node metastases, CT: computed tomography, PPV: positive predictive value, NPV: negative predictive value, ROC: receiver operating characteristic, AUC: area under the curve, CI: confidence interval.

**Table 3 cancers-15-02605-t003:** Logistic regression analysis of clinicopathological variables in relation to LNM.

	Univariate Analysis			Multivariate Analysis		
	*p* Value	Adjusted OR	95% CI	*p* Value	Adjusted OR	95% CI
**Age > 65 years ***	0.964	0.99	0.53–1.83	-	-	-
**BMI >30 kg/m^2^ ***	0.790	0.91	0.46–1.81	-	-	-
**NEEC histology**	<0.001	3.75	1.82–7.70	0.028	3.72	1.16–11.95
**CA125 > 35 U/mL**	<0.001	7.99	4.00–15.99	0.003	6.25	1.85–21.09
**CT with suspected LNM**	0.002	4.14	1.72–9.99	0.427	1.680	0.48–6.05
**Deep myometrial involvement**	<0.001	3.45	1.76 6.74	0.007	4.88	1.54–15.46
**Cervical stromal involvement**	0.009	4.21	1.42–12.50	0.018	7.87	1.42–43.78

OR: odds ratio, CI: confidence interval for adjusted OR, BMI: body mass index, NEEC: non-endometrioid endometrial carcinoma, CT: computed tomography. * = not included for multivariate analysis as *p* > 0.02.

## Data Availability

The data presented in this study are available on request from the corresponding author. The data are not publicly available due to privacy and legal reasons.

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
