# Peer review of "Preoperative CA125 Significantly Improves Risk Stratification in High-Grade Endometrial Cancer"

_cancers, 2023, doi:10.3390/cancers15092605_

Round 1

Reviewer 1 Report

This retrospective multicenter study aimed to assess whether elevated CA125 is a predictor of the spread of UC to elsewhere in the body and the survival of patients with UC.

 The topics is of interest and i would like to congratulate with Authors for their effort

My comments are:

1)   Add in the discussion section a word on the importance of evidence-based oncofertility counselling that can be adequately fulfilled for the sake of female cancer patients, in light of its complexities and multidisciplinary nature, which require thorough counselling and consent pathways. Zaami, Simona, Stark, Michael, Signore Fabrizio, Gullo Giuseppe, Marinelli Enrico. Fertility preservation in female cancer sufferers: (only) a moral obligation? European Journal of Contraception and Reproductive Health CareVolume 27, Issue 4, P335 – 3402022.

2)   Add in the discussion section a word on the importance of  Y RNA on endometrial cancer and about their potential use in tumor characterization and as therapeutic targeting to inhibit cell proliferation in oncological patients. Gulìa Caterina, Signore, Fabrizio  Gaffi, Marco, Gigli, Silvia, Votino, Raffaella, Nucciotti, Roberto, Bertacca, Luca, Zaami, Simona, Baffa, Alberto, Santini, Edoardo, Porrello, Alessandro, Piergentili, Roberto. Y RNA: An overview of their role as potential biomarkers and molecular targets in human cancers. Cancers Volume 12, Issue 5May 2020.

3)   Add in the discussion section a word on the value of Ca-125 during pregnancy as it becomes higher due to the production of decidua. Valentina D’Ambrosio, Roberto Brunelli, Lucia Musacchio, Valentina Del Negro, Flaminia Vena, Gaia Boccuzzi, Chiara Boccherini, Violante Di Donato, Maria Grazia Piccioni, Pierluigi Benedetti Panici, and Antonella Giancotti. Adnexal masses in pregnancy: an updated review on diagnosis and treatment. Tumori JournalVolume 107, Issue 1, February 2021, Pages 12-16

Author Response

Reviewer 1:

This retrospective multicenter study aimed to assess whether elevated CA125 is a predictor of the spread of UC to elsewhere in the body and the survival of patients with UC. The topics is of interest and I would like to congratulate with Authors for their effort

My comments are:

Point 1: Add in the discussion section a word on the importance of evidence-based oncofertility counselling that can be adequately fulfilled for the sake of female cancer patients, in light of its complexities and multidisciplinary nature, which require thorough counselling and consent pathways. Zaami, Simona, Stark, Michael, Signore Fabrizio, Gullo Giuseppe, Marinelli Enrico. Fertility preservation in female cancer sufferers: (only) a moral obligation? European Journal of Contraception and Reproductive Health CareVolume 27, Issue 4, P335 – 3402022.

Response 1: We thank the reviewer for the time invested in reviewing this paper and providing these useful comments. We have included the reference and a paragraph to the discussion section of the paper:

Page 10 line 290- page 11 line 296:While CA125 is a very sensitive marker that can be used in postmenopausal women in the diagnostic work-up for endometrial cancer, it might be less suitable in premenopausal women. This is due to the fact that elevated CA125 is seen in several benign and physiologically processed such as menstrual period, endometriosis, and pregnancy.

For patients with a wish of fertility preservation, evidence-based oncofertility counselling is an important part of the preoperative work-up[33]. So far, it is not clear whether preoperative CA125 in these young women can equally contribute in the risk stratification of endometrial cancer. “

Point 2: Add in the discussion section a word on the importance of  Y RNA on endometrial cancer and about their potential use in tumor characterization and as therapeutic targeting to inhibit cell proliferation in oncological patients. Gulìa Caterina, Signore, Fabrizio  Gaffi, Marco, Gigli, Silvia, Votino, Raffaella, Nucciotti, Roberto, Bertacca, Luca, Zaami, Simona, Baffa, Alberto, Santini, Edoardo, Porrello, Alessandro, Piergentili, Roberto. Y RNA: An overview of their role as potential biomarkers and molecular targets in human cancers. Cancers Volume 12, Issue 5May 2020.

Response 2: We have included the reference and a paragraph to the discussion section of the paper:

Page 11 line 315-319:In the future, other molecular and genetic markers such as non-coding RNA (ncRNA) might further stratify tumor characterization and therapeutic targeting[33, 34]. Further research will be needed to assess CA125 in relation to these markers but may enable the possibility to base the need of surgical staging and/or adjuvant treatment solely on these variables

Point 3: Add in the discussion section a word on the value of Ca-125 during pregnancy as it becomes higher due to the production of decidua. Valentina D’Ambrosio, Roberto Brunelli, Lucia Musacchio, Valentina Del Negro, Flaminia Vena, Gaia Boccuzzi, Chiara Boccherini, Violante Di Donato, Maria Grazia Piccioni, Pierluigi Benedetti Panici, and Antonella Giancotti. Adnexal masses in pregnancy: an updated review on diagnosis and treatment. Tumori JournalVolume 107, Issue 1, February 2021, Pages 12-16

Response 3: We have included the reference and a paragraph to the discussion section of the paper together with response 1.

Reviewer 2 Report

At the end of the Introduction to the manuscript, the authors articulate their goal: in women with high grade endometrial cancer who have undergone CT imaging, to determine the diagnostic value of a CA-125 test in predicting the presence of lymph node metastasis. They have performed a nice study to obtain data that bears on this question, and (in Figure 1) have presented those data. Unfortunately, though the authors have conducted many analyses that are peripheral to the question, and have not addressed the primary question in sufficient detail.

I propose that most of the narrative presented (and most of the tables) be replaced by a consideration of just those women in whom surgical staging was done, and in whom the results were conclusive. (Also, the women with distant metastases but no LNM should be excluded.) The  data that generated Figure 1 (as modified according to the above recommendation) should include the number of subjects on which the predictive values are based, and the confidence intervals around these predictive values: the small numbers of subjects suggest that such intervals might be quite wide.

These results certainly indicate that measurement of CA-125 offers additional ability to predict LNM in women with high grade endometrial cancer who have undergone imaging. But I recommend that the Discussion section address whether even the combination of imaging and CA-125 is clinically useful. In other words, is the discrimination that these tests offer adequate to forgo surgical staging in some women? That would seem to be the best use to which the results of this study could be put.

Author Response

Reviewer 2:

At the end of the Introduction to the manuscript, the authors articulate their goal: in women with high grade endometrial cancer who have undergone CT imaging, to determine the diagnostic value of a CA-125 test in predicting the presence of lymph node metastasis. They have performed a nice study to obtain data that bears on this question, and (in Figure 1) have presented those data. Unfortunately, though the authors have conducted many analyses that are peripheral to the question, and have not addressed the primary question in sufficient detail.

Point 1: I propose that most of the narrative presented (and most of the tables) be replaced by a consideration of just those women in whom surgical staging was done, and in whom the results were conclusive.

Response 1: We thank the reviewer for their extensive effort and helpful comments. We have adjusted table 2, the narrative from paragraph 3.2 to 3.4, and the narrative referring to this in the methods section. We refer to the track changes in the manuscript for these adjustments.

Point 2: (Also, the women with distant metastases but no LNM should be excluded.)

Response 2: Due to the mentioning of this important exclusion criterium we have reviewed the study cohort. Only one of the participants had distant metastases but no LNM, these were omental metastases. This participant has been excluded from the study. All other participants either had no LNM or LNM with or without distant metastases. We have included a paragraph in the method section.

Page 3 line 129-130: Patients who underwent staging with distant metastases but without LNM were excluded”.

Furthermore, Table 1, 2 and 3 and figure 1 and the text have been updated with the correct numbers. We refer to the manuscript for these updates and the new Tables and figure 1.

Point 3: The  data that generated Figure 1 (as modified according to the above recommendation) should include the number of subjects on which the predictive values are based, and the confidence intervals around these predictive values: the small numbers of subjects suggest that such intervals might be quite wide.

Response 3: The actual number of subjects have been added to Figure 1. The values shown in Figure 1 are directly derived from the study cohort and its purpose is to visually show the distribution of the cohort.

Point 4: These results certainly indicate that measurement of CA-125 offers additional ability to predict LNM in women with high grade endometrial cancer who have undergone imaging. But I recommend that the Discussion section address whether even the combination of imaging and CA-125 is clinically useful. In other words, is the discrimination that these tests offer adequate to forgo surgical staging in some women? That would seem to be the best use to which the results of this study could be put.

Response 4: We have added this very important question to the discussion.

Page 11 Line 302-306: While this study has shown that CA125 has a good predictive value for LNM and extended disease, its discriminatory value is not enough to forgo surgical staging based solely on CA125 with or without CT. The results do however confirm that CA125 is a highly valuable marker which could be used in prediction models, such as the ENDORISK model which incorporates CA125, imaging and other factors [13].”

Page 11 Line 316-319: Further research will be needed to assess CA125 in relation to these markers but may enable the possibility to base the need of surgical staging and/or adjuvant treatment solely on these variables”.

Round 2

Reviewer 2 Report

The revised manuscript has been improved, but the basic problems remain: 1. Most of the text and tables do not address the question at hand; and 2. The conclusions don’t follow from the data that ARE relevant to the question at hand.

At the end of the Introduction, the authors clearly state their goal: “We aim to determine the predictive value of CA125 with preop imaging in relation to LNM in high grade endometrial cancer”. However, the only data related to this aim appear in Figure 1 – the data presented in the tables and the text describing the tables are extraneous. Compounding the problem, the text relating to the data in Figure 1 doesn’t provide the reader with the most important features: Among women who are positive on imaging, the presence of an elevated CA125 increases the likelihood of LNM being present from 11% to 73%, BUT these %s are based on very small numbers. (That is why confidence intervals around the %s are necessary.) Among women who are negative on imaging, even those with a normal value for CA125 have a 16% chance of having LNM. The far-from-perfect discriminating ability of CA125 makes it clear that some of the adjectives used by the authors – “important” and “highly valuable” – are not appropriate.

Author Response

The revised manuscript has been improved, but the basic problems remain: 1. Most of the text and tables do not address the question at hand; and 2. The conclusions don’t follow from the data that ARE relevant to the question at hand.

Point 1: At the end of the Introduction, the authors clearly state their goal: “We aim to determine the predictive value of CA125 with preop imaging in relation to LNM in high grade endometrial cancer”. However, the only data related to this aim appear in Figure 1 – the data presented in the tables and the text describing the tables are extraneous.

Response point 1: We thank the reviewer for their effort and for these additional and important comments. We realized that our  primary and secondary aims were not clear enough to the reviewer and therefore strengthened the different research questions in the short summary, abstract and paper. Furthermore, we clarified that we evaluated both  LNM and  advanced stage, as presented by FIGO stage in table 2. As when patients are already diagnosed with advanced stage UC, there is limited benefit of detecting LNM as patients already will receive additional treatment. We hope that this important clarification better reflects the several tables and text describing the tables.

Point 2: Compounding the problem, the text relating to the data in Figure 1 doesn’t provide the reader with the most important features: Among women who are positive on imaging, the presence of an elevated CA125 increases the likelihood of LNM being present from 11% to 73%, BUT these %s are based on very small numbers. (That is why confidence intervals around the %s are necessary.) Among women who are negative on imaging, even those with a normal value for CA125 have a 16% chance of having LNM.

Response point 2: We thank the reviewer for further clarifying this important remark. We have consulted our statistician and added the confidence intervals to table 2. We agree that this information is necessary for readers to correctly interpret our results. As figure 1 serves only to visualize the population proportion within our study cohort, no confidence intervals are shown here.   

Point 3: The far-from-perfect discriminating ability of CA125 makes it clear that some of the adjectives used by the authors – “important” and “highly valuable” – are not appropriate.

Response point 3: We agree with the reviewer that clear language is important for a good presentation of study results. We have therefore changed the adjectives used to describe the value of CA125 as a predictor for LNM and advanced stage. Please refer to the manuscript for updated changes.

Round 3

Reviewer 2 Report

The manuscript is now substantially improved, due to the reframing of the question, the augmented analysis, and the use of more appropriate adjectives in describing the importance of the results.